# Optimum Volume Fraction and Inlet Temperature of an Ideal Nanoparticle for Enhanced Oil Recovery by Nanofluid Flooding in a Porous Medium

Abdullah Al-Yaari [1,*], Dennis Ling Chuan Ching [1], Hamzah Sakidin [1], Mohana Sundaram Muthuvalu [1], Mudasar Zafar [1], Yousif Alyousifi [2], Anwar Ameen Hezam Saeed [3] and Abdurrashid Haruna [1,4]

[1] Department of Fundamental and Applied Sciences, Universiti Teknologi PETRONAS, Seri Iskandar 32610, Perak, Malaysia

[2] Department of Mathematics and Statistics, Faculty of Science, Universiti Putra Malaysia, Serdang 43400, Selangor, Malaysia

[3] Department of Chemical Engineering, Universiti Teknologi PETRONAS, Seri Iskandar 32610, Perak, Malaysia

[4] Department of Chemistry, Ahmadu Bello University, Zaria 810107, Nigeria

* Correspondence: abdullah_20001447@utp.edu.my; Tel.: +60-183103931

**Abstract:** Nowadays, oil companies employ nanofluid flooding to increase oil production from oil reservoirs. Herein the present work, a multiphase flow in porous media was used to simulate oil extraction from a three-dimensional porous medium filled with oil. Interestingly, the finite element method was used to solve the nonlinear partial differential equations of continuity, energy, Darcy's law, and the transport of nanoparticles (NPs). The proposed model used nanofluids (NFs) empirical formulas for density and viscosity on NF and oil relative permeabilities and NP transport equations. The NPs thermophysical properties have been investigated and compared with their oil recovery factor (ORF) to determine the highest ORF. Different NPs ($SiO_2$, $CuO$, and $Al_2O_3$) were used as the first parameter, keeping all parameters constant. The simulation was run three times for the injected fluid using the various NPs to compare the effects on enhanced oil recovery. The second parameter, volume fraction (VF), has been modeled six times (0.5, 1, 2, 3, 4, and 5%), with all other parameters held constant. The third parameter, the injected NF inlet temperature (293.15–403.15 K), was simulated assuming that all other parameters are kept constant. The energy equation was applied to choose the inlet temperature that fits the optimum NP and VF to determine the highest ORF. Findings indicated that $SiO_2$ shows the best ORF compared to the other NPs. Remarkably, $SiO_2$ has the lowest density and highest thermal capacity. The optimum VF of $SiO_2$ was 4%, increasing the ORF but reduced when the VF was higher than 4%. The ORF was improved when the viscosity and density of the oil decreased by increasing the injected inlet temperature. Furthermore, the results indicated that the highest ORF of 37% was obtained at 353.15 K when $SiO_2$ was used at a VF of 4%. At the same time, the lowest recovery is obtained when a volume of 5% was used at 403.15 K.

**Keywords:** flooding; porous media; optimum volume fraction; temperature inlet; mathematical model; enhanced oil recovery





## 1. Introduction

A standard oil production method from an oil reservoir porous medium (PM) is flooding, in which water flooding or water with surfactants or polymer additives displaces oil from a PM. Several recently published papers propose NF flooding as a promising agent for enhancing ORF. An NF consists of NP with a base fluid, usually water, and the size, VF, and type of NP can vary the physical properties of the NF significantly. The main physical properties of NPs are dynamic viscosity, density, wettability, thermal capacity, and interfacial tension [1–3].

NPs have already been successfully implemented in medical, electrical, material science, and other sectors [4,5]. The oil and gas energy sectors are beginning to conduct many

lab experiments and numerical models to study the potential of NPs for EOR during the water flooding process. The findings provided an outstanding performance for improving oil production aimed at either the capillary-trapped oil or the passing of oil due to the many potential features of NPs. Among the many benefits of NPs in EOR applications is their flexibility under different pressures and high temperatures. NP's unique mechanical and thermal properties can change the base fluid's interfacial characteristics and thermal conductivity. Therefore, NPs can enhance ORF techniques and are environmentally beneficial and cost-effective [6–8].

Numerous techniques, including chemical and thermal procedures, have been developed recently to improve oil recovery. Polymer and surfactant flooding is a typical chemical procedure widely employed, particularly in China. Polymers are commonly utilized to increase the injected nanofluids' viscosity. The capillary force increased, and interfacial tension was reduced by flooding with surfactants, which could move trapped oil outside the PM and lead to EOR. Despite these benefits, two disadvantages may impact the future acceptability of polymer. First, it can be affected by reservoir conditions, making it less effective as an EOR, especially in formations with high temperatures and saltiness. Lastly, this type of polymer may harm the environment [9,10].

Flooding, in which water or NF is used to extract oil from porous media, is a standard method for extracting oil from terrigenous reservoirs. In several recent studies, NF flooding has been recommended as a potential agent for EOR instead of water flooding. Only 5% to 15% of oil can be extracted by primary recovery, and EOR is referred to enhance oil extraction from the reservoir medium. An NF is a fluid (typically water) that has NPs added to it. The physical characteristics of the NF might vary substantially according to the NP type and VF. The viscosity and density of the water would change if NPs were added to the water. NF flooding would improve the PM sweep efficiency by reducing the fingering effect, interfacial tension, and a change from oil-wetted to water-wetted. The NFs decrease oil viscosity and experimental results revealed that flooding with NFs boosts the ORF relative to flooding with water. The most often used NPs in EOR are silicon, aluminum, and copper oxide NPs [11,12].

The following properties make NPs suitable for EOR: The NP's tiny size enables optimal transit into the reservoir's PM without blocking the pore throats, increasing the injected fluid's viscosity and enhancing the flood's mobility ratio, leading to increased sweep efficiency and more effective EOR; the excellent thermal properties of NPs make for rapid heat transfer from the injected fluid to oil with less density and viscosity, making it easier to flow outside the reservoir PM [13,14].

The use of different NPs to enhance ORF during nano flooding has been the subject of most studies. NP-based waterflooding is studied using a combination of lab experiments and numerical simulations. NF flooding has the potential to be more cost-effective than nearly all other EOR methods. The addition of NPs would change and optimize the base fluid TPs. Subsequently, obtaining the optimal mobility ratio allows the NF flooding qualities to be controlled for improving the ORF during the EOR process. The ability of NPs to form a net structure with the help of hydrogen raises the viscosity of the NF that is injected, which changes the NF shear stress. AlamiNia et al. demonstrated that even at low shear rates, the NF viscosity is greater than the water viscosity [15]. Depending on the reservoir PM, temperature, specific NPs, and their optimal concentrations are essential for improving the efficacy of NF flooding [16,17]. So, it is crucial to look into the factors affecting NF flooding. The kind of NPs dispersed in the base fluid, concentration, size, and NF injection temperature are essential parameters for the ORF [18,19]. In other words, choosing a suitable NP is a significant factor. In the literature, it has been reported that $SiO_2$, $TiO_2$, CuO, and $Al_2O_3$ NPs can be used for EOR.

Several analytical approaches have been used to model and analyze oil flows in porous media. One commonly used approach is Darcy's law, which states that the flow rate of a fluid through a porous medium is directly proportional to the pressure gradient. This relationship has been used to predict oil flow through a given porous medium and to design

and optimize production systems. The Darcy-Brinkman equation corrects the empirical Laws of Darcy by relating the average flow velocity to the applied pressure gradient, the fluid viscosity, and the permeability of the solid media [20].

Many recent publications use the Brinkman equation instead of the classic Darcy's law to represent flows through porous media. media [21]. These studies are concerned with the physical applications of the equation and the mathematical formalism of analytical and numerical solutions. The Darcy–Brinkman equation was estimated near those encountered in oil extraction situations. The results showed that the Darcy formulations fit Darcy–Brinkman equations for their clarity and accuracy [22]. Another analytical approach is the use of pore-scale models. These models simulate the behavior of fluids at the pore scale and are used to understand the fundamental mechanisms that govern fluid flow in porous media. Pore-scale models can be used to predict the behavior of fluids under different conditions and provide insight into the physical properties of the porous medium. In a broader sense, the equation can be seen as a continuous model that describes how two media interact when one is solid and the other is a fluid that can not be compressed [23].

Ju et al. conducted one of the first studies to simulate NF flow in PM. In constructing their model, transfer equations of NPs were employed to evaluate the accumulation rate of NPs due to NP trapping in the PM [24]. The model given by Ju et al. is an extension of the NPs equation transport in PM based on the formulation proposed by Liu and Civan, which is modeling for NP migration in PM [25]. This model quantitatively calculates the changes in the relative effectiveness of permeability of water and oil phases and the change in the ORF following injection of NF. This model has been utilized to determine the distribution of NP concentrations (VF) and the medium's porosity and absolute permeability. At the end of the study, Ju et al. [24] concluded that the ORF in the environment has improved with the help of NPs, even though the NPs have made the PM less permeable. Considering capillary forces, Brownian motion, and neglecting the wetting, El-Amin et al. expanded Ju et al.'s technique to a two-dimensional model in the PM for EOR by using NF flooding [26]. Feng et al. performed numerical studies on how parameters such as the length of time and the VF of NPs affect the efficiency of ORF based on El-Amin et al.'s model [27]. Still, their model did not consider the capillary pressure and variations in wettability. Sepehri et al. examined variations in the rock wettability since the NPs were added to the base fluid using the Darcy model [28]. Yu et al. modeled the flow of NFs in PM and found that the water's salinity extensively influences the migration of NPs [29].

Darcy's equation and the conservation equations of mass, momentum, and energy could well be utilized to solve the numerical modeling of NF flow in porous media during oil displacement. The physical properties of NF in each control volume are calculated based on the concentration of NPs. Gharibshahi et al., using a two-dimensional microscopic model, investigated and examined the effect of wettability on oil displacement, the impact of NP type, concentration, and size, as well as fluid injection temperature [30–32]. Most experimental studies have found the NF flooding approach with these properties is commonly employed in EOR research. Few studies have been conducted to model NF flooding in three-dimensional PM for EOR.

Water flooding is the most practical and widely used oil recovery method. However, this method cannot recover all the oil inside the reservoirs due to capillary forces. Because of this, much research has been done to find new EOR agents that can be added to injection water to reduce trapped oil and boost the ORF. With progress in nanoscience, researchers found that adding NPs to the water improved flooding performance and increased oil production. Researchers have examined the feasibility of using NPs for EOR in controlled lab settings and computer-based simulations in the past decade. Most studies have found that NPs are a suitable addition to the injected fluids for EOR applications [33,34]. The NP types as new and different solutions have been looked into for choosing the ideal NP for more oil production with less time and cost. A study of NPs was conducted by Ahmadi et al., who compared $SiO_2$ to $Al_2O_3$ and MgO. The data indicated that the $SiO_2$ NP had the highest ORF [35]. Ogolo et al. studied the effects of aluminum oxides, silicon

dioxide, nickel, and iron on EOR. They discovered that dispersing aluminum oxide NPs in brine and distilled water significantly increases oil recovery [36]. Ogolo et al. investigated several NPs that regulate fine migration in formations. The primary NP size in their study ranged from 10 to 70 nm, and the NPs were dispersed in three different base fluids. They discovered that aluminum oxide NP improved oil recovery more than the other NPs [36]. The researchers found that some NPs increased the ORF, but the rest negatively affected the EOR. So, the type of NP is a crucial parameter for EOR.

Significant research has been conducted to identify the optimum VF to be added to the injection base fluid to increase the ORF. These studies indicated that adding NPs to the injected fluid up to 4% would increase the ORF, but the ORF is starting to decrease for amounts higher than 4% of the NPs. By injecting NF, which is made up of the based fluid (water) and a 4% nanoparticle VF, the highest final ORF that can be reached is 25% [37]. Therefore, the NP VF is an essential parameter in the EOR process for increasing the ORF. The VF percentage has to be used with caution because some of the percentages negatively affect the EOR process and cause a decrease in the ORF. Variations in NF thermophysical properties (TPs) are considered in the two-phase modeling of NPs in the PM [38]. Heat transfer equations have been used to determine the best NF inlet temperature for the phases' TPs (NF and oil) to increase ORF. Table 1 summarizes some of the details of NF flooding experiments and simulations.

**Table 1.** Summary of the NF flooding experiments and simulations.

| References | NPs | Parameters | Extra Oil Produced (%) |
|---|---|---|---|
| Alomair et al. [39] | $Al_2O_3$, $SiO_2$ $NiO$, $TiO_2$ | NP type | −16.94 to 23.72 |
| Ragab et al. [40] | $SiO_2$, $Al_2O_3$ | NP type and VF | $SiO_2$ (8.74 to 13.88) $Al_2O_3$ (−8.12 to −4.65) |
| Hendraningrat et al. [41] | $SiO_2$ | NP VF, PV | 5.93 to14.29 |
| Li et al. [42] | $SiO_2$ | NP type and VF | 5 to 15 |
| Maghzi et al. [43] | $SiO_2$ | NP VF | 8.7 to 26 |
| Hendraningrat et al. [44] | $SiO_2$, $Al_2O_3$, $TiO_2$ | NP type | 7 to 11 |
| Hendraningrat et al. [45] | $SiO_2$ | permeability NP VF | 0 to 8.41 |
| Kazemzadeh et al. [46] | $SiO_2$, $NiO$, $Fe_3O_4$ | NP type and VF | $SiO_2$ (22.6) $NiO$ (14.6) $Fe_3O_4$ (8.1) |
| Ragab et al. [47] | $SiO_2$ | NP size and VF | 5 to 10 |
| EI-Diasty [48] | $SiO_2$ | NP size and VF | 9 to 19 |
| Ogolo et al. [36] | $SiO_2$ | Base fluid and NP | 13.3 to 24.1 |
| Ehtesabi et al. [49] | $TiO2$ | NP VF | 10 to 14 |
| Haroun et al. [50] | $CuO$, $NiO$, $Fe_3O_4$ | NP type | $Fe_3O_4$ (8.19) $NiO$ (7.59) $CuO$ (14.07) |

Simulations of oil displacement from reservoirs modeled using nanofluid flooding processes, concentrating on the change in wettability of the medium caused by nanoparticle accumulation during the nanofluid injection. But the change in wetting properties represents one of the proposed mechanisms to explain the effect of NF injection on increasing oil production. In addition, some EOR models have variant wettability values giving rise to the same or different ORF distribution. So, the changes in the wettability properties

are just one of the proposed mechanisms of NF injection to enhance oil recovery [51]. The parameters that change in NF thermophysical properties have not been studied sufficiently in the two-phase simulation in three-dimensional PM. In this study, we proposed a model for EOR which considered the TPs by employing their empirical correlations on the relative permeability (RP) formulas of the NF, oil, and NP mass transfer, for the first time. Furthermore, the EOR simulation utilizes NF flooding processes by considering the type of NPs, volume fraction, and the injected fluid inlet temperature. Different NPs, volume fractions, and intake temperatures (293.15 and 403.15 K) were investigated to determine the optimum NP, volume fraction, and injected inlet temperature for EOR. Lastly, this research aims to conduct a three-dimensional model that can simulate NF flooding in PM for EOR.

## 2. Numerical Implementation

Using computer simulations, such as fluid flow in PM, complex systems can be studied using computational fluid dynamics (CFD). CFD is a method for studying fluid flow, heat transfer, and associated side effects such as chemical reactions. This method uses computer simulations to study these processes. While experimental studies can provide valuable information about a process, CFD can be used to examine critical and unique situations, minimize reaction time and research expenses, and acquire thorough and detailed knowledge of the process. FEM was used in the present modeling, a standard numerical method for CFD studies. COMSOL Multiphysics software, whose main structure is entirely based on FEM, is used for the simulation in the present study. The FEM examines numerical methodologies for nanofluid flow in PM for EOR. There is much work in the literature on CFD and FEM, especially in the EOR process in PM modeling [52–54]. CFD modeling is used in this study to assess the NF flooding process's effects on the ORF's rise. A three-dimensional geometry was generated using COMSOL Multiphysics software, considering the changes in porosity, permeability, capillary, and the different types and VFs of NPs in the base fluid (water). TPs were examined to find the optimal VF for EOR. In the following, the hypotheses, governing equations used in the COMSOL Multiphysics software code to simulate the nanofluid flooding process into the porous medium, and the input parameters of the model have been discussed:

### 2.1. The Input Parameters and Mathematical Model Flow Chart

In this study, the FEM was used to solve nonlinear equations such as the continuity and Darcy equations for NF and oil. The velocity and pressure fields are replaced in the saturation equations to determine NF and oil saturation. The capillary pressure and RP of NF and oil are calculated using the saturation equations after resolving Darcy's and the continuity equations. An implicit method was employed in the first step of this research to solve the Darcy, continuity, energy, and concentration equations. After that, the equations were solved using an explicit approach. The revised porosity and absolute permeability have been then computed for the following stage, which had been changed due to NP deposition. Also, NP deposition at the end was considered to determine the optimum NP concentration feasible for the EOR. Table 2 shows the input parameters used in the proposed model for the EOR process, and Figure 1 depicts the flow chart of the mathematical solution.

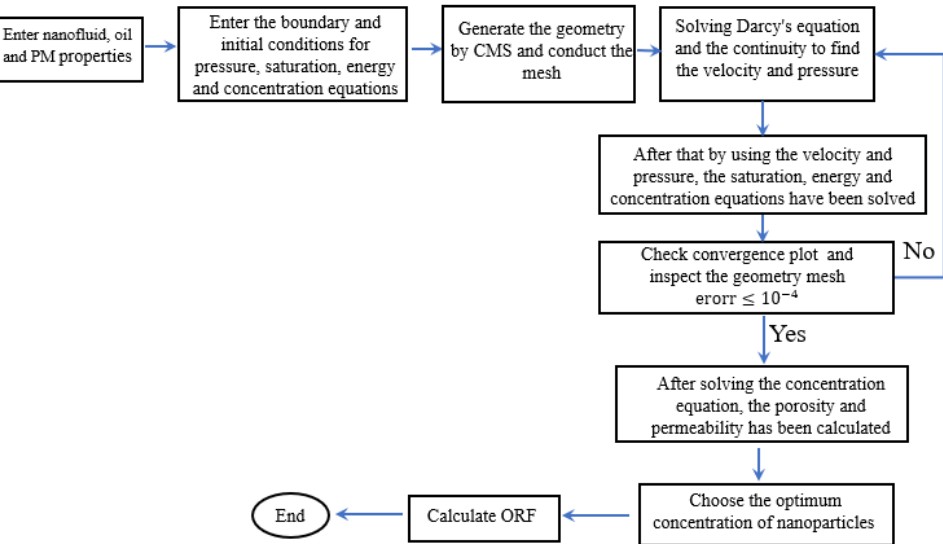

**Figure 1.** The flow chart of the mathematical model.

**Table 2.** Model parameters for the simulation of NF flooding in the PM for EOR.

| Parameter | Value | Unit | Description |
|---|---|---|---|
| $T_{init}$ | 293.15 | K | The initial temperature of the system |
| $T_{in}$ | 350 | K | The inlet temperature of NF |
| $p_{nf}$ | 0 | Pa | NF initial pressure |
| $p_o$ | 0 | Pa | Oil outlet Pressure |
| V | 2 | m$^3$ | Volume of geometry |
| A | 2 | m$^3$ | Cross section of the PM |
| $S_o$ | 1 | - | The initial oil saturation in the PM |
| $S_{nf}$ | 1 | - | Inlet saturation of NF |
| B$_c$ | 2 | Pa | Inversely proportional to the square root of permeability |
| $\varepsilon_0$ | 0.2 | - | Initial porosity of the PM |
| K$_0$ | $10^{-9}$ | m$^3$ | Initial absolute permeability |
| $\rho_w$ | 990 | kg/m$^3$ | Water density |
| $\rho_o$ | 880 | kg/m$^3$ | Oil density |
| $\rho_s$ | 2714 | kg/m$^3$ | Solid matrix density |
| $C_{p,p}$ | 851 | J/kg·K | Solid matrix-specific heat |
| k$_s$ | 2.2 | W/(m·K) | Solid matrix thermal conductivity |
| $\rho_{SiO_2}$ | 2220 | kg/m$^3$ | Silicon density |
| $\rho_{Al_2O_3}$ | 3970 | kg/m$^3$ | Aluminum density |
| $\rho_{CuO}$ | 6310 | kg/m$^3$ | Copper density |
| $C_{p,SiO_2}$ | 745 | J/(kg·K) | Silicon specific heat |
| $C_{p,Al_2O_3}$ | 765 | J/(kg·K) | Aluminum specific heat |
| $C_{p,CuO}$ | 531 | J/(kg·K) | Copper specific heat |
| $k_{SiO_2}$ | 36 | W/(m·K) | Silicon thermal conductivity |
| $k_{Al_2O_3}$ | 40 | W/(m·K) | Silicon thermal conductivity |
| $k_{CuO}$ | 20 | W/(m·K) | Silicon thermal conductivity |
| $\phi$ | (1, 2, 4, 5) | % | NP VF |
| d$_p$ | 40 | nm | NPs diameter |
| MN$_p$ | 60 | g/mol | Molecular weight |
| $C_{p,o}$ | 1670 | J/(kg·K) | Oil specific heat |
| $\kappa_o$ | 0.13 | W/(m·K) | Oil thermal conductivity |
| $\mu_o$ | $4.5 \times 10^{-4}$ | Pa·s | Oil viscosity |
| $C_{p,w}$ | 4200 | J/(kg·K) | Water specific heat |
| $\kappa_w$ | 0.6 | W/(m·K) | Water thermal conductivity |
| $\mu_w$ | 0.001 | Pa·s | Water viscosity |

### 2.2. Geometry

In this research, light oil has been applied to evaluate the NF's effect on the ORF. In earlier research, NFs composed of water with NPs (silicon, aluminum, or copper) were the most common NFs used to increase the ORF. Table 3 shows the properties of the model's geometry created using CMS. Figure 2 depicts the geometry used in the current study, and the problem's boundary conditions are summarized in Table 4. Because the NF flooding is moving through the PM, the NF layer closest to the surface has zero velocity. The typical triangular mesh in CMS worked well enough for fluid flow problems in homogeneous PM, as illustrated in Figure 3.

**Table 3.** PM properties.

| Properties | Values |
|---|---|
| Cross-section area ($m^2$) | $1 \times 2$ |
| Width (m) | 2 |
| Height (m) | 1 |
| Depth (m) | 1 |
| Porosity (%) | 0.2 |
| Horizontal permeability ($m^2$) | $1 \times 10^{-9}$ |
| Initial water saturation (%) | 0 |

**Table 4.** Boundary conditions.

| | | | |
|---|---|---|---|
| The rectangular prism inlet side | $S_{nf} = 1$ | $\mathrm{n} \cdot \rho_{nf} u_{nf} = 0.001$ m/s | $T = T_{in}$ |
| The rectangular prism outlet side | $p_o = 0$ Pa | $-\mathrm{n} \cdot \rho_o u_o = \rho_o u_{o0}$ | $-\mathrm{n} \cdot k_{\text{eff}} \nabla T = 0$ |
| All surfaces except the inlet and outlet | | $-\mathrm{n} \cdot \rho_o u_o = 0$ | $-\mathrm{n} \cdot k_{\text{eff}} \nabla T = 0$ |
| Initial condition | $S_o = 1$ | $p_{nf} = 0$ Pa | $T_{init} = 293.15$ K |

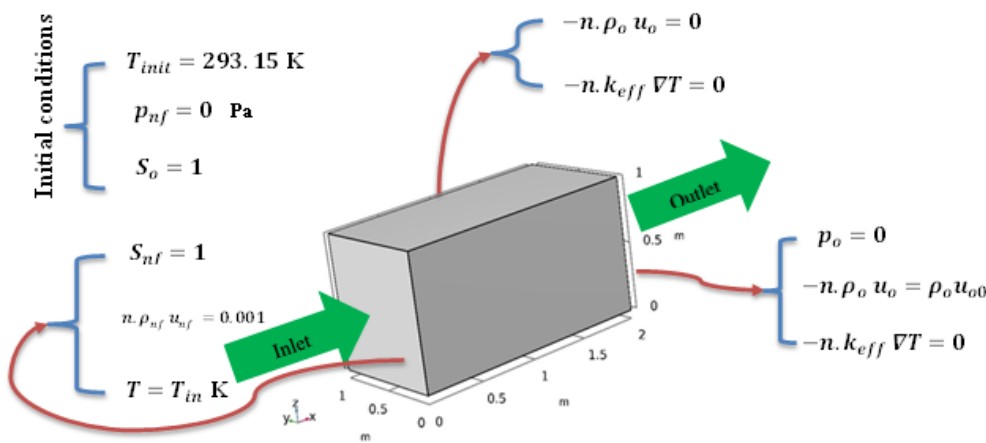

**Figure 2.** The model geometry.

### 2.3. Governing Equations

The NF flooding was investigated initially as water flooding, considering that the VF equals zero to examine the system's mathematical model in the oil reservoir. Increased oil recovery has been achieved by using NFs in place of water. The saturated oil ($S_o$) and NF saturation ($S_{nf}$) in the PM should be established to determine the ORF. The following sections have discussed the governing equations of NF and oil flow in PM.

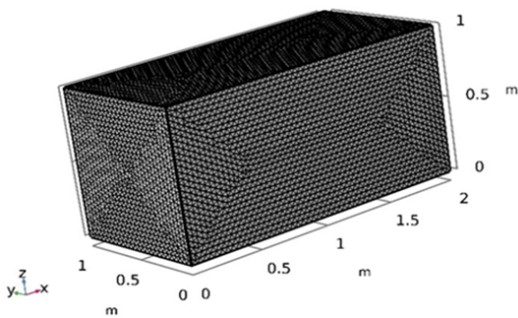

**Figure 3.** Mesh geometry.

### 2.3.1. Nanofluid-Oil Flow in the Porous Medium

Darcy's law and mass conservation equations govern two-phase water-oil flow in porous media, assuming no mass transfer occurs between the two phases. First, the PM was saturated with oil and had zero NF saturation, and then the NF was injected through the PM's left side at the velocity $(u_{nf0})$ of 0.001 m/s, displacing oil through the right side. The velocity of oil $u_o$ on the left side was zero.

$$-\text{n} \cdot \rho_o u_o = 0 \text{ m/s} \tag{1}$$

$$-\text{n} \cdot \rho_{nf} u_{nf} = 0.001 \text{ m/s} \tag{2}$$

The NF pressure is set to zero Pa at the outlet:

$$p = p_{nf} = 0 \text{ Pa} \tag{3}$$

In both time and space, the total velocity of water and oil $u = u_{nf} + u_o$ was constant.

$$u_{nf} = -K m_{nf} \nabla p \tag{4}$$

$$u_o = -K m_o \nabla p \tag{5}$$

where $u_{nf}$ and $u_o$ are the velocities of NF and oil, respectively, and the mobilities of NF and oil are $m_{nf}$ and $m_o$, respectively, and K is the absolute permeability.

$$m_{nf} = k_{rnf} / \mu_{nf} \tag{6}$$

$$m_o = k_{ro} / \mu_o \tag{7}$$

the functions $k_{nf}$ and $k_o$ are the water and oil RP, respectively,

$$\nabla \cdot \left( \rho_{nf} u_{nf} \right) = -\varepsilon \rho_{nf} \frac{\partial S_{nf}}{\partial t} \tag{8}$$

$$\nabla \cdot \left( \rho_o u_o \right) = -\varepsilon \rho_o \frac{\partial S_o}{\partial t} \tag{9}$$

where the saturation of NF and oil are $S_{nf}$ and $S_o$, and the summation of NF and oil saturation equals one [55].

$$S_{nf} + S_o = 1 \tag{10}$$

Equation (11) is found by adding together Equations (8) and (9).

$$\nabla \cdot \left( u_{nf} + u_o \right) = 0 \tag{11}$$

then

$$\boldsymbol{u} = \boldsymbol{u}_{nf} + \boldsymbol{u}_o = \boldsymbol{u}_{inj} = 0.001 \text{ m/s} \tag{12}$$

The entire velocity is indicated by $\boldsymbol{u}$. Equations (4) and (5) are combined, and after that, Equation (12) is utilized to generate Equation (13).

$$\boldsymbol{u} = -\text{K}\left(m_{nf} + m_o\right)\nabla p \tag{13}$$

Resolving Equation (4) for $\nabla p$ and substitute in Equation (13) led to Equation (14).

$$\boldsymbol{u}_{nf} = \left(m_{nf}\boldsymbol{u}\right)/\left(m_{nf} + m_o\right) \tag{14}$$

Equations (8) and (12) created Equation (15), which is a function of NF saturation ($S_{nf}$) since the ($\boldsymbol{u}$) is the velocity of injected NF.

$$\nabla \cdot \left(\rho_{nf}\left(\left(m_{nf}\boldsymbol{u}\right)/\left(m_{nf} + m_o\right)\right)\right) = -\varepsilon\rho_{nf}\frac{\partial S_{nf}}{\partial t} \tag{15}$$

Equations (16) and (17) are utilized to obtain the NF density and viscosity, respectively, and $\varnothing$ the NP VF [56]:

$$\rho_{nf} = \varnothing\rho_{np} + (1 - \varnothing)\rho_w \tag{16}$$

$$\mu_{nf} = \mu_w\left(1 + 39.11\varnothing + 533.9\varnothing^2\right) \tag{17}$$

The effective saturation ($S_e$) is employed in the relative permeabilities to calculate the ORF:

$$S_e = \frac{S_{nf} - S_{rnf}}{1 - S_{rnf} - S_{ro}} \tag{18}$$

where ($S_{rnf}$) and ($S_{ro}$) are the residual NF saturation and oil saturation, respectively.

### 2.3.2. Nanofluid and Oil Relative Permeabilities

Due to changes from oil-wet to water-wet and nanoparticle retention in a porous medium, the relative permeability correlations may be extended to mixed-wet systems. Equations (19) and (20) describes the NF and oil RP, respectively [57]:

$$k_{rnf} = 3S_e + \frac{2}{\lambda} \tag{19}$$

$$k_{ro} = \left(1 - S_e^2\right) \times \left(1 - S_e^{1 + \frac{2}{\lambda}}\right) \tag{20}$$

where $\lambda$ is the pore size distribution index.

### 2.3.3. Capillary Pressure

The injection of NF into the oil reservoir PM is intended to change the medium's wettability from oil-wet to water-wet. As a result, it is worthwhile to include a more general correlation for capillary pressure in the model to describe its variations as wettability changes. The model calculates the diffusion between the NF and water by knowing the capillary pressure. Here are two models that were utilized in the simulation:

$$P_c = P_{nf} - P_o \tag{21}$$

And $P_c$ is the capillary pressure inside the oil reservoir PM.

$$P_c = -B_c \log(S_e) \tag{22}$$

where the coefficient $B_c$ is inversely proportional to the square root of permeability. The capillary pressure for the Brooks-Corey model may be calculated using Equation (23):

$$P_c = P_{ec} S_e^{\frac{-1}{\lambda}} \tag{23}$$

The entrance capillary pressure is $P_{ec}$, and $\lambda$ is the pore size distribution index in the equation above.

### 2.3.4. Nanoparticle Transport

The flow of nanoparticles is based on the behavior of nanoparticles in water. If nanoparticles remain in the water, nanoparticle movement is only a one-phase equation and is considered Brownian diffusion. In this work, only one interval size of the nanoparticles spread in water was assumed in the transport equation. The NFs should be used at the highest NP concentration possible; otherwise, the efficiency will be significantly lower than with water flooding. Equation (24) defines the NP transfer partial differential equation [58,59].

$$u\frac{\partial C_i}{\partial t} + \varepsilon S_w \frac{\partial C_i}{\partial t} - \frac{\partial}{\partial x}\left(\varepsilon S_w D_c \frac{\partial C_i}{\partial x}\right) + R_i = 0 \tag{24}$$

$$D_c = \frac{k_{rw}}{\mu_w}k(S_w - 1)\frac{\partial P_c}{\partial s_w} \tag{25}$$

The following equations are the initial and boundary conditions:

$$t = 0 \rightarrow C_{\text{init}} = 0 \tag{26}$$

$$X = 0 \rightarrow C = C_{\text{in}} \tag{27}$$

The NP concentration is $C_i$, $D_c$ is the diffusion, and $R_i$ is the NP rate loss in the path. Since the first injection was done with water, there are no NPs. So, the initial condition of NP concentration is zero. By using Ju and Dai's model [60], $R_i$ in Equation (28) refers to the amount of NPs loss that may be divided into two sections:

- The amount of surface is connected to the aqueous phase of all NPs.
- The NPs are stuck in the pore media route, which blocks the flow path.

$$R_i = \frac{\partial \sigma}{\partial t} + \frac{\partial \sigma^*}{\partial t} \tag{28}$$

where $\sigma^*$ is the volume of trapped NPs in pore throats per unit bulk volume, and $\sigma$ is the volume of NPs deposed on pore surfaces per unit bulk volume.

$$\frac{\partial \sigma}{\partial t} = k_d u C \tag{29}$$

$$\frac{\partial \sigma^*}{\partial t} = k_p u C \tag{30}$$

where the initial conditions

$$\sigma = \sigma^* = 0 \tag{31}$$

where $k_d$ and $k_p$ are constant coefficients for interface deposit and NP entrainment, respectively, and $u$ is the essential NP phase velocity of 0.00025 m/s.

The porosity and permeability of the PM might be reduced by the deposition of NPs on the pore surface and the obstruction of the pore throat. The following equation may be used to calculate the porosity of NPs during transport:

$$\varepsilon = \varepsilon_0 - (\sigma + \sigma^*) \tag{32}$$

where

$$K = K_0 \left[ (1 - F)\alpha_f + \frac{F\varepsilon}{\varepsilon_0} \right]^n \tag{33}$$

where $\varepsilon_0$ and $K_0$ represent the initial permeability and porosity, and K and $\varepsilon$ represent the PM's permeability and porosity, respectively. $\alpha_f$ is the constant for fluid seepage permitted by blocked pores. At the same time, $n$ is an empirical coefficient between 1 to 3, $F$ represents the percentage of the initial cross-section area open to the flow, and has a value ranging from 0 to 1. Ju et al. found that the constant values for $\alpha_f$, n, and $F$ are 0.26, 2.5, and 0.6, respectively, compared to what had been published before.

### 2.3.5. Nanofluid Inlet Temperature

When the NF inlet temperature increased, the oil temperature in the PM increased due to the heat transfer rate from the NF to oil being higher than that between water and oil. Adding NPs increased thermal conductivity, allowing more effective heat transfer from NFs to oil. The flooding method would profit from two reasons as temperatures rise. First, the oil's viscosity was reduced, so its sticky and weight were reduced. Second, as temperatures rose, the density of oil decreased. This made oil lighter, so getting it out of reservoirs took less energy. This section introduces the following thermophysical properties that were combined from NF and oil, as shown in Table 5, and used in the energy equations inside PM [61]:

$$\left(\rho_{tot} C_p\right)_{\text{eff}} \frac{\partial T}{\partial t} + \rho_{tot} C_p \boldsymbol{u} \cdot \nabla T + \nabla . q = 0 \tag{34}$$

$$q = -k_{\text{eff}} \nabla T \tag{35}$$

$$\left(\rho C_p\right)_{\text{eff}} = \theta_{\text{p}} \rho_p C_{p,p} + \left(1 - \theta_{\text{p}}\right)\rho C_p \tag{36}$$

$$k_{\text{eff}} = \theta_{\text{p}} k_s + \left(1 - \theta_{\text{p}}\right)k_{tot} \tag{37}$$

In the above equations, $\theta_{\text{p}}$ represents the porous rock VF, defined as $1 - \varepsilon_0$. The initial temperature in the system is 293.15 K. The heat equation boundary condition is defined as Equation (38):

$$-nq = \rho \Delta H un, \quad \Delta H = \int_{T_{\text{in}}}^{T} C_p dT \tag{38}$$

where $T_{in}$ is the temperature of the input NF for determining the ORF, for the PM outlet is the following condition.

**Table 5.** Properties that were used in the energy equation.

| NF Properties | Combining NF and Oil | Unit |
|---|---|---|
| Density ($\rho_{tot}$) [62] | $\rho_{tot} = S_{nf}\rho_{nf} + \left(1 - S_{nf}\right)\rho_o$ | kg/m$^3$ |
| Specific heat capacity ($C_{p,tot}$) [63] | $C_{p,tot} = S_{nf}Cp_{nf} + \left(1 - S_{nf}\right)Cp_o$ | J/(kg·K) |
| Thermal conductivity coefficient ($k_{tot}$) [64,65] | $k_{tot} = S_{nf}k_{nf} + \left(1 - S_{nf}\right)k_o$ | W/(m·K) |

### 2.3.6. Assumptions

- The NF consists of NPs and water, considered homogeneous, meaning the NP and water are considered one phase.
- The PM is homogeneous.
- The NF and PM are an incompressible flow as it is always in the reality PM.
- Two-phase Darcy's law was used for modeling NFs and oil flows in the PM.
- The PM is assumed to be isothermal.

### 2.4. Grid Independency

Five meshes with different grid cells (600,000, 700,000, 800,000, 900,000, and 1,000,000) were generated to investigate grid independence. The model was run with fixed variables and a different number of cells each time to compare the five values of NF saturation in the PM and select the optimal mesh number of cells. The VF of NP $SiO_2$ was 1%, and the viscosity of the base fluid (water), oil, and NF were set equal to 0.001, $4.5 \times 10^{-4}$, and 9.8 Pa·s, respectively. The water, oil, and NF density were set to 990, 880, and 1100 kg/m$^3$, respectively. For modeling, the displacing NF's velocity at the inlet was set to 0.001 m/s, and the oil pressure was zero at the outlet. The PM was filled with oil, and there was no water inside the PM at first. The residual oil saturation was 0.3%, and the NF saturation distribution in the computational domain was examined. Table 6 displays the mesh with different cells and the ORF as a function of a specific point (1 m, 0.5 m, 0.5 m, 200 s). As shown, mesh number 3 is the best mesh used in all simulations. In addition, CMS has a mesh plot that reveals how good the quality of the mesh cells is. Cells are best when their colors are green, which means the cells have the highest quality. Figure 4 shows that the meshing quality of the current study geometry is outstanding since all cells are green with one value. Furthermore, Figure 4b shows the convergence plot, which is time-domain convergence for NF saturation and provides a solution to the system of equations at a given period of 250 s. Figure 4c explains the solution with an error smaller than $10^{-5}$.

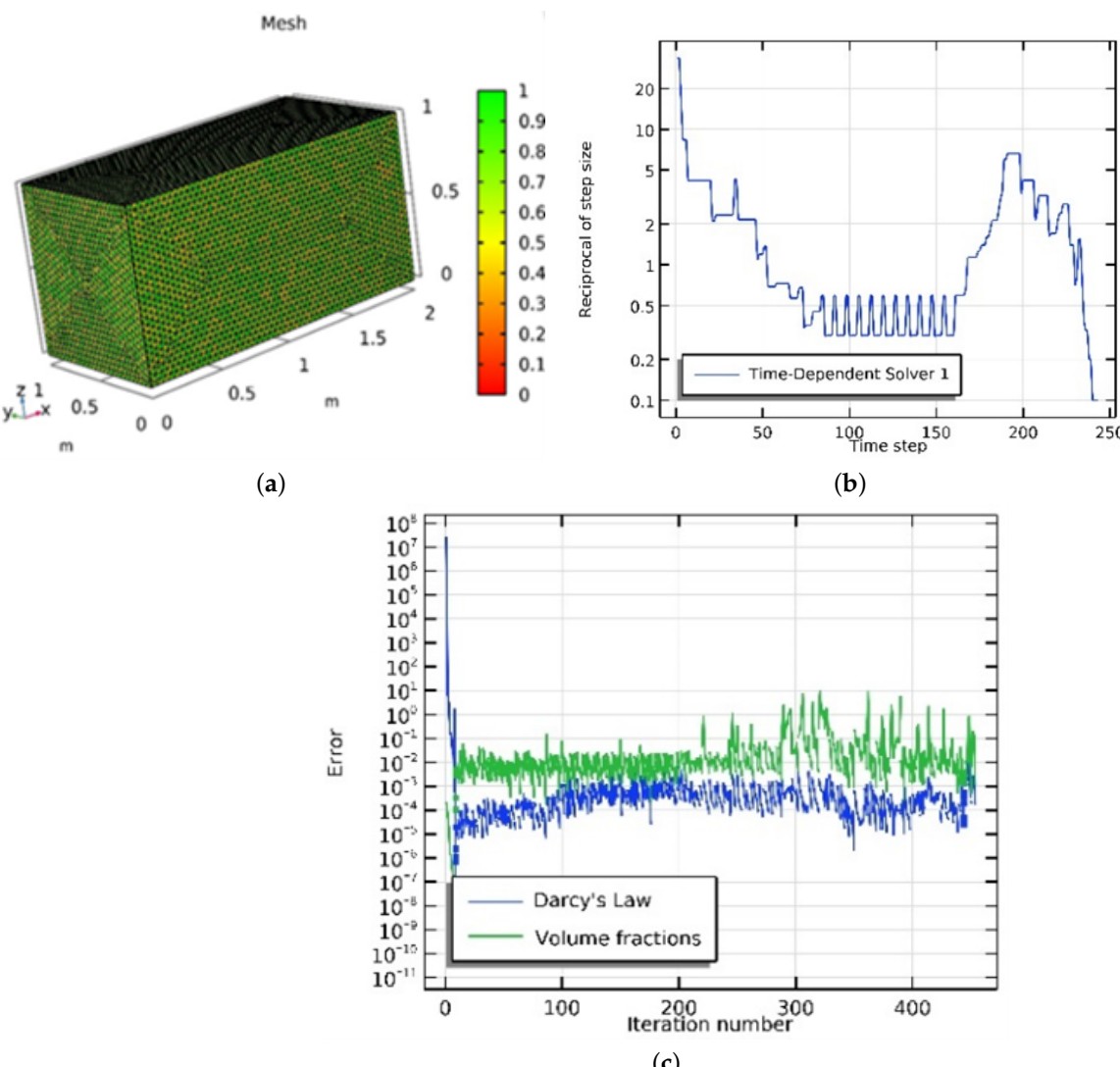

**Figure 4.** (**a**) Inspection of the mesh, (**b**) convergence plot, and (**c**) the error less than $1 \times 10^{-5}$.

**Table 6.** ORF at (1 m, 0.5 m, 0.5 m, 200 s) with different grids.

| No. | No. of Cells | ORF |
|---|---|---|
| 1 | 300,000 | 0.575 |
| 2 | 400,000 | 0.626 |
| 3 | 500,000 | 0.632 |
| 4 | 600,000 | 0.634 |
| 5 | 7,000,000 | 0.635 |

*2.5. Verification*

The injected NF into the reservoir to displace the oil has been modeled. This study presents three-dimensional modeling whose structure is similar to reservoir PM. The numerical findings of the NF injection into the PM have been compared to the experimental data from Minakov et al. [19]. The injected NF in the experiment was $SiO_2$ into the PM. The model's geometry in their investigation is a 3D PM with a porosity of 0.25, an absolute permeability of $5.2 \times 10^{-20}$ m$^2$, and a size of $4.836 \times 10^{-15}$ m$^3$. As shown in Figure 5, the experimental and numerical results agree well. The ORF showed an average relative error of 1% when comparing numerical and experimental data.

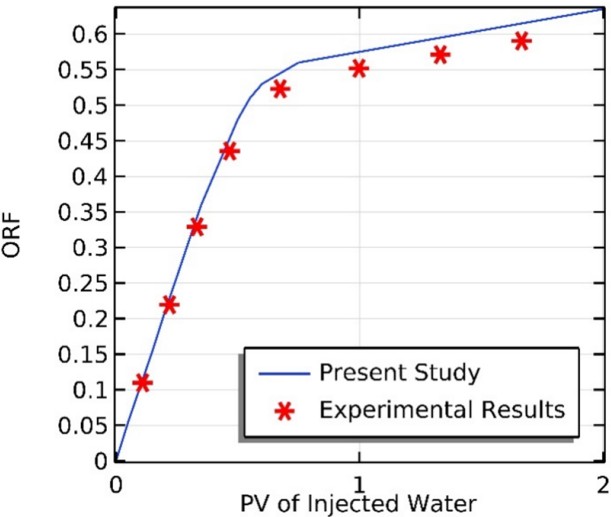

**Figure 5.** Comparison of numerical and experimental findings.

## 3. Results and Discussion

After the numerical and experimental results were in good agreement, the effects of several parameters, such as the NP types and VF, the injected NF thermophysical properties, and inlet temperature on EOR, were studied. The effects of the different injected NF VF into the PM on the saturation and ORF for the duration of the NF flooding process for five different VF (0.5, 1, 2, 3, 4, 5%) were examined on the changes in the NF thermophysical properties. The density and viscosity of all NFs increased as the VF increased, as shown in Figure 6. However, it was concluded that NF (water-$SiO_2$) had the lowest density and the highest thermal capacity of all NFs. Due to the strength of intermolecular forces, viscosity could increase as the VF rises. The NP viscosity performs an essential title role in the mobility ratio of the base fluid (water) since it increases the injected water viscosity, leading to the EOR. Ranjbar et al. showed that adding NPs to the base fluid increased the viscosity. This may cause longer fluid retention in the oil, making it easier to get oil out of the PM [66]. Minakov et al. conclude that ORF decreased because the oil viscosity and NF viscosity ratio decreased. So, if the oil viscosity was fixed and the NF viscosity increased, the ratio would decrease, causing a reduction in oil production [19]. Therefore, the optimum VF is essential for EOR.

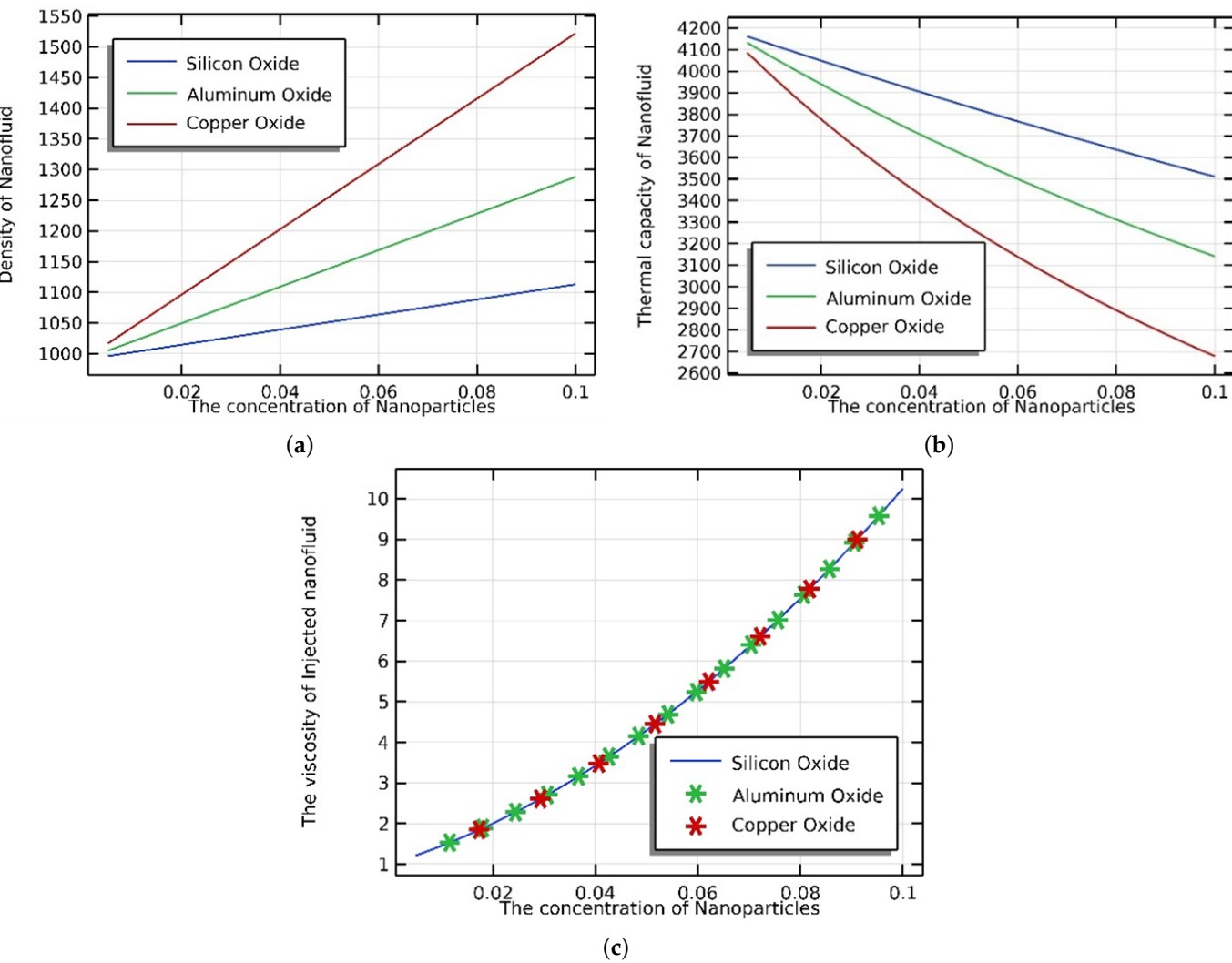

**Figure 6.** (**a**) The effect of NP volume fraction on density, (**b**) on thermal capacity, and (**c**) on viscosity.

The NPs are crucial for the EOR application because adding different NP types to the injected base fluid into the PM of the oil reservoir increases the ORF. So, nanotechnology has become a practical approach to oil production. The present study has investigated the NFs' flooding effects on ORF compared to the base fluid (water). The simulation process using the proposed model is run many times in COMSOL Multiphysics software using different NPs with water for selecting the best type of NF for EOR under oil reservoir PM conditions (higher pressure, temperature, and water salinity). The inlet temperature of NF is one of the essential parameters for EOR. Increasing the temperature led to increasing the ORF, but over a specific value, this temperature started to influence the EOR negatively, where the ORF began to decrease. So, the optimum inlet temperature, the best NP, and its optimum volume fraction are essential to ensure that more oil is extracted quickly and cheaply.

*3.1. The Nanofluid's Effect on EOR*

The NPs CuO, SiO$_2$, and Al$_2$O$_3$ were compared in this research to choose the best NP for EOR, where these NPs were mixed with water separately to create the NF (water-CuO, water-SiO$_2$, and water-Al$_2$O$_3$). Then each NF was injected into the PM to simulate the EOR process. The NF characteristics were extracted by plugging the numerical values of the density and viscosity of the NPs and water into the NF empirical equations in Table 7. The special effects of adding different NPs on the TPs of NF and NF saturation were

investigated. Also, for all three NFs, the impact of a VF ranging from 1% to 5% on their TPs and saturation was studied.

### 3.1.1. The Impact of the Nanofluid's Density and Viscosity on EOR

This study investigated different NFs such as water-SiO$_2$, water-Al$_2$O$_3$, and water-CuO. The changes in the density of the NFs due to the nanoparticle's VF are shown in Figure 6a. This picture shows that the density of all NFs (water-CuO, water-SiO$_2$, and water-Al$_2$O$_3$) increased when the VF increased. The NF (water-SiO$_2$) density is the lowest compared to water-CuO and water-Al$_2$O$_3$, and the NF (Water-CuO) had the highest density compared to the others due to the highest density of CuO. Figure 6b shows how increasing NP VF in base fluid (water) affects thermal capacity. As the value of the VF rises, the thermal capacity decreases. The greatest thermal capacity of the various NFs (water-CuO, water-SiO$_2$, and water-Al$_2$O$_3$) was found to be SiO$_2$ in the water-based. The NF density and the thermal capacity of the NPs affect the thermal capacity. In addition, the influence of VF on the SiO$_2$–water, Al$_2$O$_3$–water, and CuO–water viscosities were measured. The NPs (SiO$_2$, Al$_2$O$_3$, and CuO) with different VFs were added to water, and the viscosity measurements were made using the empirical correlation of the NF viscosity in Table 7. Figure 6c shows the NFs' viscosities increasing as the NPs VF increases. The empirical correlation of the NF density has been used in the NF and oil RP equation since the NF based on water with silicon oxide is the lowest, making the RP of NF flooding with silicon bigger than the others. Simultaneously, the RP of oil is lower than the other oils when the simulation is run with aluminum or copper, which in this case, led to EOR. So, it is more precise and accessible than other models to predict the NP's higher ORF than other NPs.

**Table 7.** NF thermophysical Properties.

| NF Properties | Equation (Combining NP and Water) | Unit |
|---|---|---|
| Density ($\rho_{nf}$) [62] | $\rho_{nf} = \phi\rho_{np} + (1 - \phi)\rho_w$ | kg/m$^3$ |
| Specific heat capacity ($C_{P_{nf}}$) [67] | $C_{P_{nf}} = (\phi\rho_{np}C_{P_{np}} + (1 - \phi)\rho_w C_{Pw})/\rho_{nf}$ | J/(kg·K) |
| Viscosity ($\mu_{nf}$) [68,69] | $\mu_{nf} = \mu_w(1 + 39.11\phi + 533.9\phi^2)$ | Pa·s |

### 3.1.2. The Impact of the NPs' Mass Transfer on EOR

Using NFs to increase ORF and improve flooding function has recently become very common. The NFs should be used at the highest Nanoparticle VF possible; otherwise, the efficiency is significantly reduced compared to water flooding. Darcy's law and the continuity equations have been included in the governing equation section for oil and NF during flooding. Because the porous media is wholly saturated with various fluids, the total saturations must equal one. NP can lose mass in two ways when it moves through PM with pore bodies and throats: by settling on the surface of pores and by blocking the pores' throats. But Liu and Civan have added the net rate of NP loss (R) to their new model. The PM porosity and permeability could be reduced if NPs were deposited on the pores' surface or if the throats of pores were blocked. The Ju and Fan equation was applied to evaluate NP transportation.

### 3.2. The Effects of Nanofluid Thermophysical Properties on EOR

This study investigated different NFs, such as water-SiO$_2$, water-Al$_2$O$_3$, and water-CuO. The changes in the density and viscosity of the NFs due to the VF of the nanoparticles were investigated. The density and viscosity of all NFs (water-CuO, water-SiO$_2$, and water-Al$_2$O$_3$) increased when the VF increased. The density of NF (water-SiO$_2$) was the lowest compared to water-CuO and water-Al$_2$O$_3$. An increase in VF in the base fluid (water) affects thermal capacity. As the value increases, the thermal capacity decreases, and the largest thermal capacity of the different NFs (water-CuO, water-SiO$_2$, and water-Al$_2$O$_3$) was found for SiO$_2$ in the water base. The empirical correlation of NF density was used in the NF and oil RP equation because the NF density is lowest based on water with silicon

oxide, which makes the RP of NF flooding with silicon larger than the others. At the same time, the RP of oil is lower than the other oils when the simulation is performed with aluminum or copper, which resulted in EOR in this case.

Figure 7 demonstrates that, over a time range of 0–400 s, the oil from the PM displaced by the injection of water–copper oxide is less than the oil displaced by water–silicon and water–aluminum oxide. So, it is observed that silicon oxide NPs are much better than aluminum and copper oxide. Silicon oxide's density is less than the others, and its thermal capacity is greater than the other NPs, as shown in Figure 6a,b. In addition, aluminum oxide's density is less than copper oxide's density, and its thermal capacity is more significant than copper oxide's thermal capacity, as shown in Figure 6a,b. So, the NF (water-NP) with lower density and higher thermal capacity is better for EOR. Silicon is better than aluminum and copper. Simultaneously, aluminum oxide is better than copper oxide, proven by the saturation equation, as shown in Figure 7. Therefore, the TPs can determine the best NPs for EOR. Esfe and Esfandeh investigated the TPs changes in the injected NF (base fluid-NPs) for developing the model of the EOR process [51].

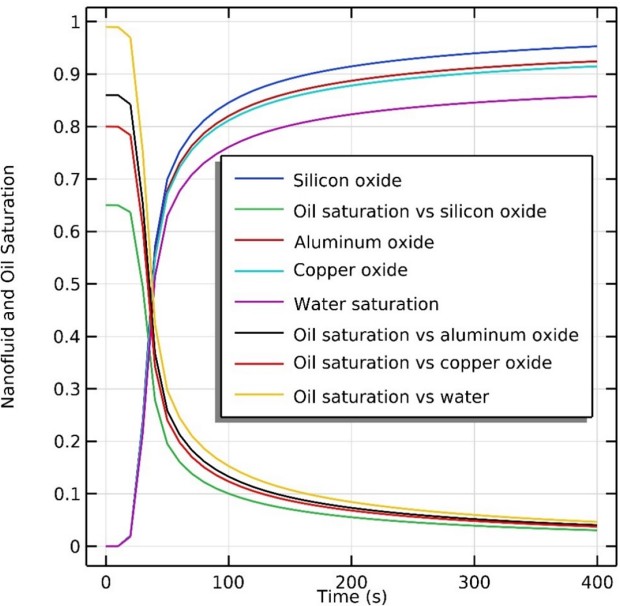

**Figure 7.** Saturation of nanofluid and oil versus time.

Figure 7 shows the saturation of NFs; water, water-$SiO_2$, water-$Al_2O_3$, and water-CuO against the oil saturation inside the oil reservoir PM. It was observed that the oil saturation inside the PM was lowest when the PM had been injected with the NF (water-$SiO_2$). The oil saturation was highest when the water was injected into the PM without NP. So, the best NF is water-$SiO_2$ for producing oil from oil reservoir PM, and after that, water-$Al_2O_3$, water-CuO, and water, respectively. So, as shown in the diagram for the 100–400th-second injection range, the NF flooding parameter increases and forces the oil to flow outside the PM via the outlet boundary.

Esfe et al. investigated three commonly used NPs with base fluid (water), which are water-$SiO_2$, water-$Al_2O_3$, and water-CuO. In their study, the effects of these three NPs were examined on the ORF, and it was discovered that $SiO_2$ showed nearly a 1% better recovery rate than $Al_2O_3$ and CuO [70]. Hendraningrat and Torsaeter evaluated the ORF and discovered that $TiO_2$ has the highest ORF than $Al_2O_3$ and $SiO_2$. Their study results are consistent with the present measurement, where $TiO_2$ (titanium) with the base fluid (water) has the lowest density and highest thermal capacity. Hence, TPs play an essential role in the oil displacement mechanism for EOR [44].

To identify the best candidates among the other NPs reported in the literature that can be used in the petroleum industry, Nazari et al. examined various NPs with the

base fluid water. The eight NFs: water-ZrO$_2$, water-CaCO$_3$, water-TiO$_2$, water-SiO$_2$, water-MgO, water-Al$_2$O$_3$, water-CeO$_2$, and water-CNT have been examined and injected into the PM twice; the first time with only water and the second time with NPs. After preliminary evaluation by contact angle measurements, water-CaCO$_3$ and water-SiO$_2$ have been discovered to have the highest oil recovery of the other NFs examined by core flooding experiments. The present study agrees well with their results, where water-CaCO$_3$ and water-SiO$_2$ have the lowest density and highest thermal capacity than others [71].

### 3.3. The Effects of Adding Nanoparticles to Water on EOR

The 3D pattern of the PM's saturated water and oil phases at 100, 200, 300, and 400 s after the flooding process is shown in Figure 8a–d, where the red and blue colors denote the water and oil saturation, respectively. Figure 8d shows that the water flooding in the PM at 400 s is more significant than the water saturation at 100, 200, and 300 s, i.e., the oil displacement by water at 400 s is greater than the oil production in the other cases. Then, over time, the oil production increased until the residual oil could not be forced outside the PM by water flooding and time. Figures 8a–d and 9a–d show the difference between water and NF flooding at different times (100, 200, 300, and 400 s). The saturation of NF in 100, 200, 300, and 400 s is always higher than the water saturation (100, 200, 300, and 400 s), respectively. That means more oil production in NF flooding, and the oil is leached more efficiently from the PM when using NF. In comparison to the results of previous studies, many studies demonstrated that injecting NF flooding of a base fluid into an oil reservoir represents a novel chemical EOR method. The NF flooding can recover approximately fifty percent of unrecoverable in-place oil reserves through primary and secondary recovery [72,73].

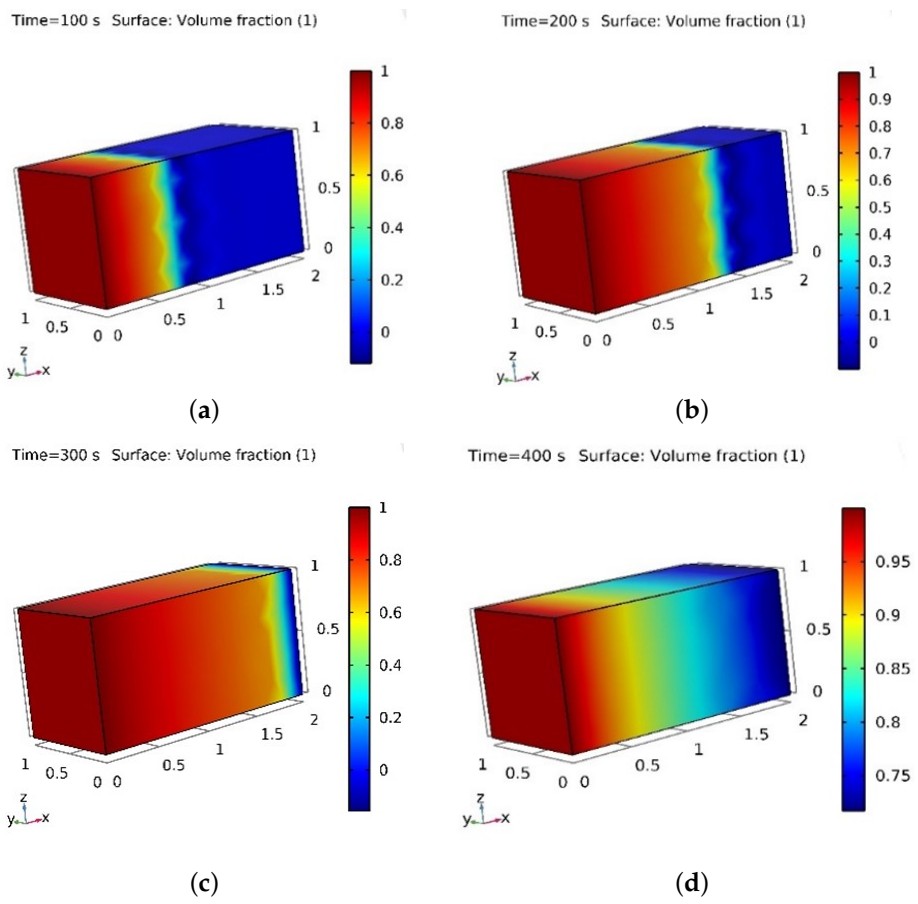

**Figure 8.** (**a**) Saturation of water at 100 s, (**b**) 200 s, (**c**) 300 s, and (**d**) 400 s.

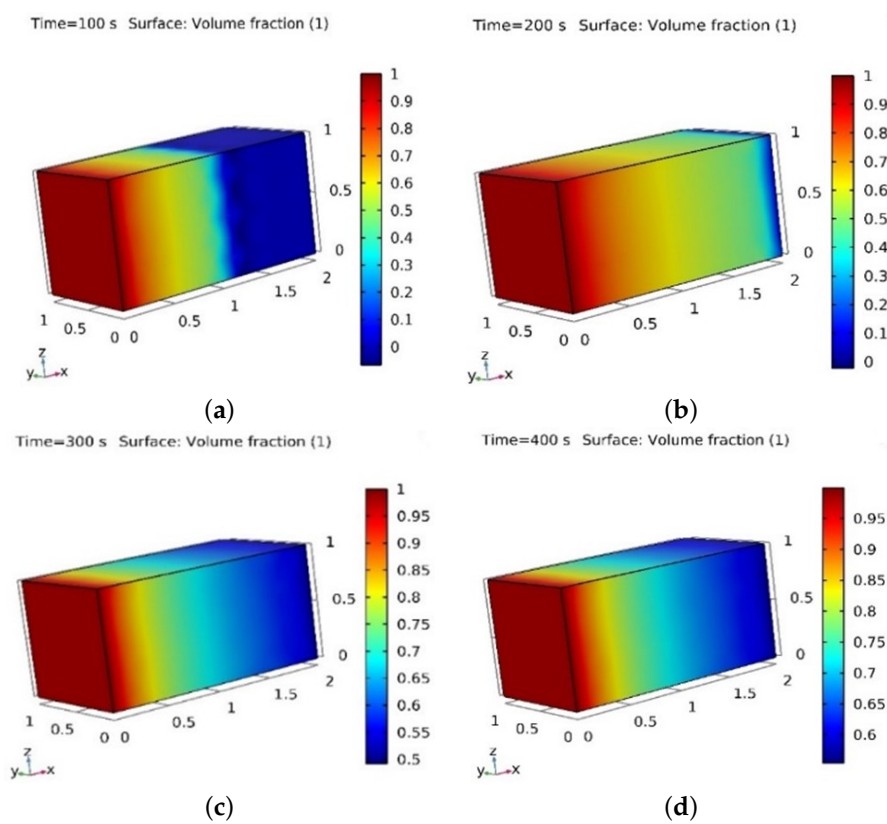

**Figure 9.** (**a**) Saturation of NF at 100 s, (**b**) 200 s, (**c**) 300 s, and (**d**) 400 s.

### 3.4. The Effects of Nanoparticle Types on EOR

Figure 10a–d shows that the oil is displaced by water flooding at a lower level than the other NFs: water-silicon, water-aluminum, and water-copper, at 100 s of the injection time, where the red and blue colors are the injected fluid and the oil, respectively. Compared to each other at 100 s, it is observed that water-silicon saturation is higher than water-aluminum and water-copper saturation inside the PM, which means more oil production by the NF water-silicon oxide. In contrast, the water-aluminum saturation at 100 s led to more oil production than the injected water-copper flooding into the PM. In their research, Ramezanpour and Siavashi concluded that NF (water-silicon oxide) flooding could significantly enhance ORF compared to the water flooding case [74]. Kazemzadeh et al. discovered that a new NF made from $SiO_2$ and $Fe_3O_4$ added to the base fluid (water) increased oil recovery differently depending on which NPs were used in the PM [75].

### 3.5. The Effects of Nanofluid Inlet Temperature on EOR

An inlet temperature was raised from 293.15 K to 403.15 K to investigate the injected base fluid temperature as a parameter to determine which inlet temperature fits and is optimal with the injected NF for improving oil production. This is because the oil saturation starts to decrease when the temperature is over 353.15 K, and there is no significant difference in the temperature between 293.15 K and 403.15 K. So, the NF saturation in 3D geometry with an inlet temperature of 293.15 K and 353.15 K at a 100 s flooding process in the PM was compared. As shown in Figures 10a–d and 11a–d, respectively, the NF saturation at 293.15 K for water, water-silicon, water-aluminum, and water-copper at 100 s is lower than the NF saturation at 353.15 K for water, water-silicon, water-aluminum, and water-copper at 100 s, respectively. So, more oil is extracted at 353.15 K than at 293.15 K. The higher temperature boosts the EOR process, but when the temperature is over 353.13, the ORF starts to decline, as shown in the following results. Therefore, the optimum temperature parameter is essential for the EOR process.

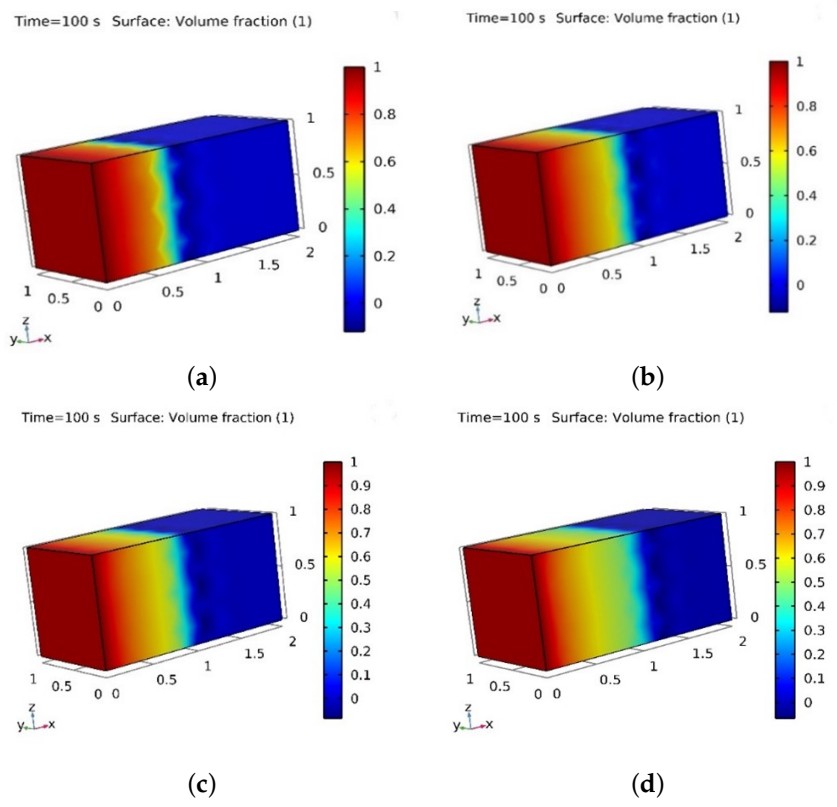

**Figure 10.** (**a**) Water saturation at 100 s, (**b**) copper at 100 s, (**c**) aluminum at 100 s, and (**d**) silicon oxide at 100 s.

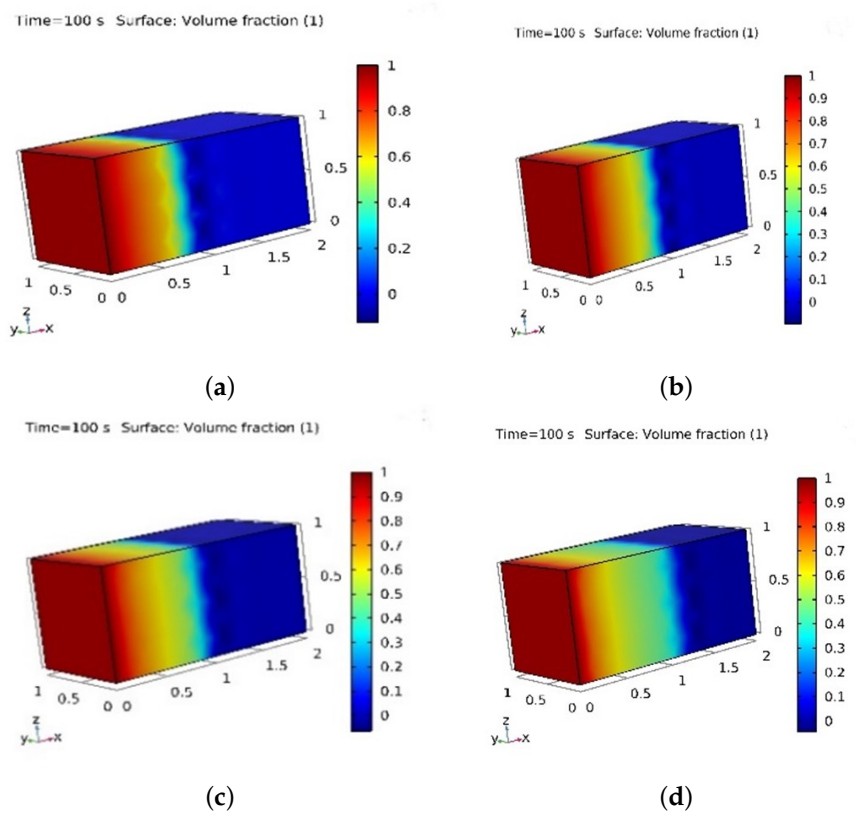

**Figure 11.** (**a**) Water saturation at 100 s with $T_{in} = 353.15$ K, (**b**) copper at 100 s with $T_{in} = 353.15$ K, (**c**) aluminum at 100 s with $T_{in} = 353.15$ K, and (**d**) silicon oxide at 100 s with $T_{in} = 353.15$ K.

### 3.6. The Pressure and Velocity Distribution Inside the Oil Reservoir PM Changed during the Nanofluid Flooding

The proposed model has been used to find the NF saturation at a range of 0–400 s using the pressure and velocity boundary and initial conditions for the NF and oil at the geometry inlet and outlet. Figure 12a–c shows Darcy's law Equation (4) has been used to find the pressure distribution inside the oil reservoir PM at 100, 200, 300, and 400 s. The pressure inside the oil reservoir PM can be found using Darcy's law which depends on the velocity and the constant absolute permeability. Because the NF and oil pressure at the inlet and outlet were both set to zero and the NF velocity at the inlet was 0.001 m/s, which is very slow, it was noticed that the pressure for 400 s was slightly higher than the pressure distribution at 100, 200, and 300 s. The initial and boundary conditions of the model must be consistent with the reality of the oil reservoir PM because, as is customary in the EOR process, the pressure inside the PM is low. Al-Yaari et al. demonstrated that the quantity of pressure in the PM did not change considerably over time due to the slow NF velocity inside the PM [38].

The oil and NF velocity inside the oil reservoir PM is very slow, and it is considered a laminar flow because the Reynolds number is less than the unit. So, the velocity distribution for 400 s inside the PM during the EOR process is approximately the same as for 100, 200, 300, and 400 s, as shown in Figure 13a–c. After getting the NF pressure, the NF and oil velocity distribution have been obtained from Equations (4) and (5).

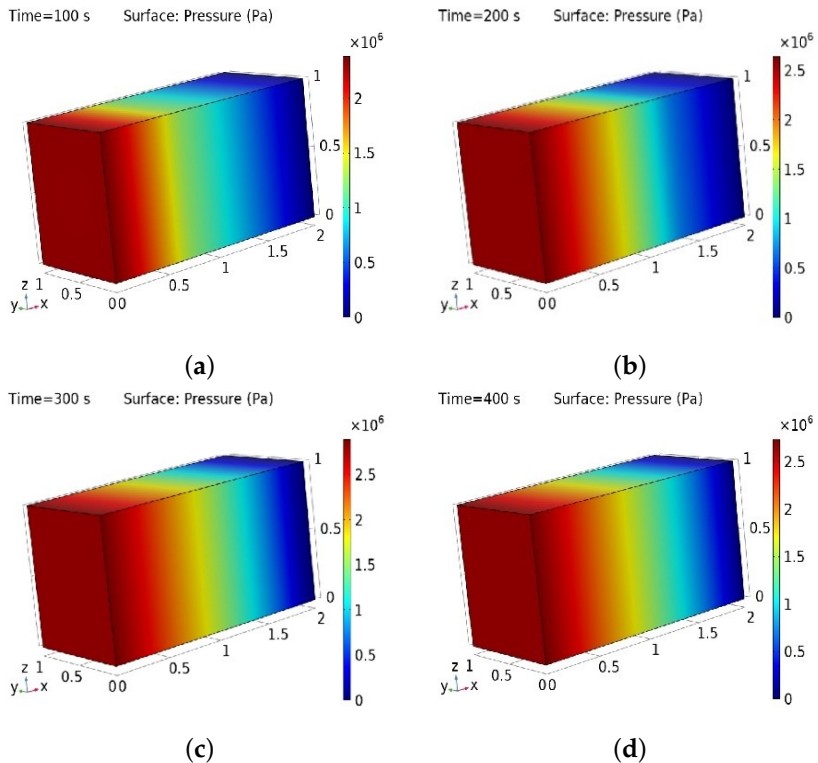

(a)   (b)   (c)   (d)

**Figure 12.** Pressure at (**a**) 100 s, (**b**) 200 s, (**c**) 300 s, and (**d**) 400 s.

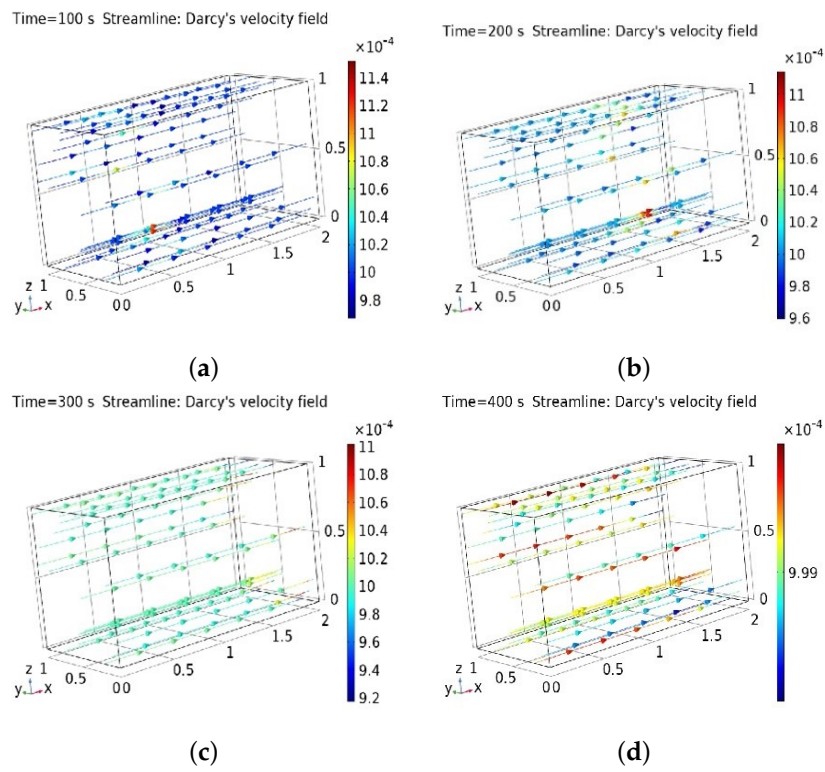

**Figure 13.** Darcy's velocity of NF inside the PM at (**a**) 100 s, (**b**) 200 s, (**c**) 300 s, and (**d**) 400 s.

*3.7. Volume Fraction Effects on NF and Oil Relative Permeabilities*

As shown in Figure 14a, the RP of the NF and water inside the PM increases with increasing VF. According to Equations (4) and (5), NF saturation and RP are directly related; as the NF saturation inside the PM increases, the NF RP also increases. In addition, the V is crucial for improving the NF RP and decreasing the oil RP. So, we can say that increasing the VF parameter positively affected oil extraction from PM in the range of vol 0–4% and had a negative impact at vol 5% because the RP of the NF started to decrease. Figure 14b depicts the RP of the oil within the PM at various VFs. As shown in the graph, as the VF parameter increases, the oil saturation decreases since the oil's RP is proportional to its saturation and is inversely proportional to the NF saturation phase inside the PM. So, when the VF parameter was increased, the RP of the oil inside the PM decreased in the range of 0–4%, leading to more oil production, but when the VF was between 4% and 5%, it led to a decrease in oil production because in this period the oil RP increased. Therefore, the optimum NPs VF in the present study for the RP of NF and oil is approximately 4%,

In their experiment, Lu et al. showed a significant increase in water saturation after adding the NPs to the base fluid. Lu et al. shows that the oil and NF phases' RP curves decreased and increased, respectively. So, the NPs improved the oil phase's ability to flow in the low permeability of the oil reservoir PM [76]. This means a good agreement with the present study results.

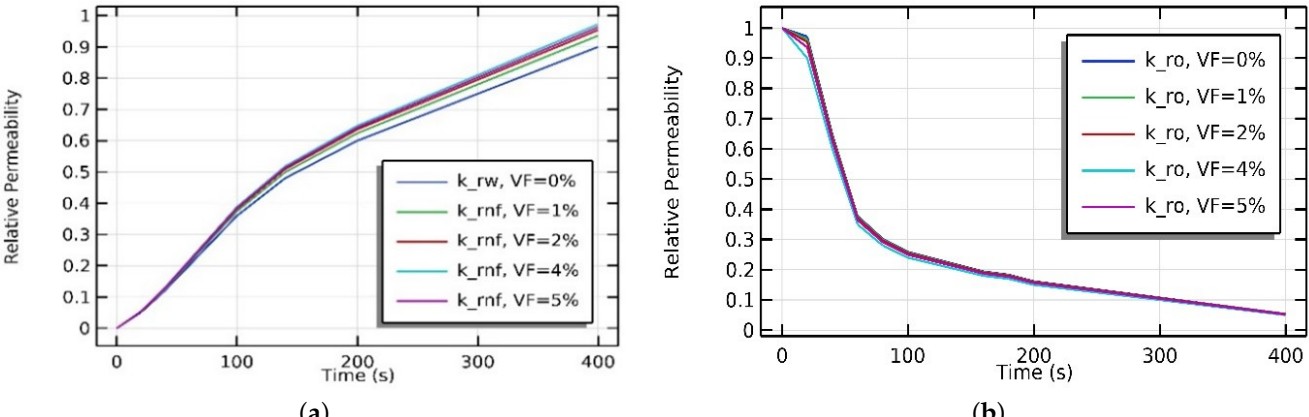

**Figure 14.** The relative permeabilities of nanofluid with different nanoparticle VFs (0, 1, 2, 4 and 5%) of and (**b**) The relative permeabilities of oil with different nanoparticle VFs (0, 1, 2, 4 and 5%).

*3.8. Nanoparticle Types and Pore Throat Effects on the Oil Recovery Factor*

The model has been run four times for water-SiO$_2$, water-Al$_2$O$_3$, and water-NiO; NFs were examined to determine the best NP for increasing ORF more than the water flooding process. As shown in Figure 15a, all NFs' ORF curves are more significant than the water ORF curve, indicating that the NF water-SiO$_2$ produced the highest ORF compared to the other NPs and 18% more than water flooding. So, using the NPs increases the ORF and improves oil production. This is because the NPs have TPs that are important for changing the properties of the base fluid, such as viscosity, density, and thermal capacity. Because of the density variation between water and NPs, they agglomerate at the entrance of pores with a minor throat. Because of the accumulation, the injecting fluid can flow towards adjacent pores, increasing their pressure. This causes the oil in neighboring pores to move and produce more oil. The oil displacement reduces pressure and leads to the gradual restoration of pore blockages. Pore channel blocking can occur when the diameter of injected NPs is larger than the pore throats they flow through. So, before these NFs are injected with the base fluid into the oil reservoirs, the NPs size should be evaluated to ensure that the NPs block pores with a small throat to get more oil production from the ground, as shown in Figure 15b. When compared to the other NFs, it is clear that water-silicon oxide increases the ORF better than water-aluminum oxide and water-copper oxide. This is because NP silicon oxide has the lowest density and highest thermal capacity than the others, its efficiency under reservoir conditions (high pressure, temperature, and water salinity), and its ability to resist asphaltene precipitation. So, this NF is anticipated to be the ideal type of NF flooding for EOR. Buckley and Fan, in their experiment, showed that the change in wettability and the decrease in the effect of capillary pressure could make silica NFs produce more oil [77]. Hendraningrat et al. concluded that the VF of NPs blocked tiny pore throats because these particles stick together around the pores, resulting in the NFs flowing to other pores with wide throats causing more oil production [78].

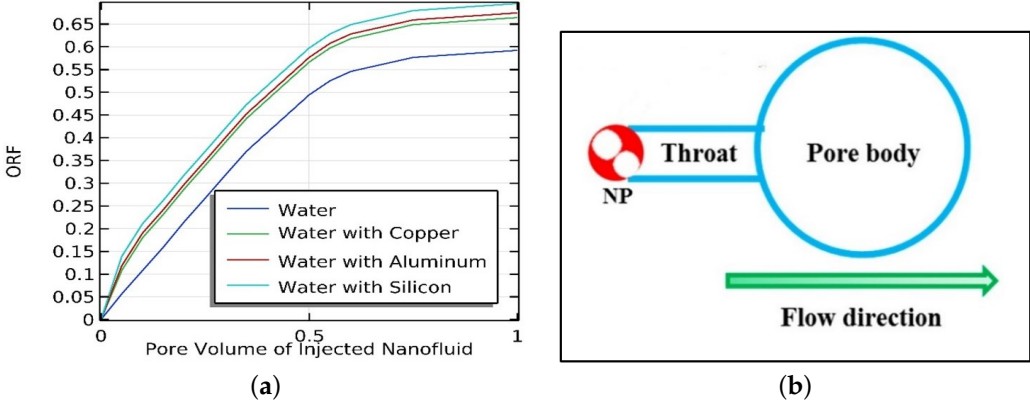

(**a**) (**b**)

**Figure 15.** (**a**) ORFs for PV of water with different NPs of vol. 4% at 293.15 K, (**b**) NPs block the pore small throat for EOR.

### 3.9. The Effect of Nanoparticle Volume Fractions on the Porosity, Absolute Permeability, and ORF

Notably, increasing the NP VF to 4% led to an increase in the ORF, as shown in Figure 16, due to the NF viscosity and density increases, and the thermal capacity decreased in the range of vol. 0–4%. Nevertheless, no significant changes in the ORF were observed in the range of vol. 4–4.2%. When the NPs VF increased from 4.2% to 5%, the ORF decreased because the adsorption reduced the porosity and absolute permeability of the PM, as shown in Figure 17a,b. So, the optimum VF to improve the ORF could be 4.0%. Maghzi et al. examined the effect of NPs' VF in NF (water with silica) and discovered that 3.5% is the optimal NP volume fraction. When the VF of an NP is greater than 3.5%, the ORF slope decreases, as determined by an analysis of the ORF during NF (water with silica) injection conducted by Ju et al. In addition, Dezfuli et al. concluded that the ORF slope decreases between 4.5 and 5% and that the optimal VF of NPs for EOR could be 3.5% [37].

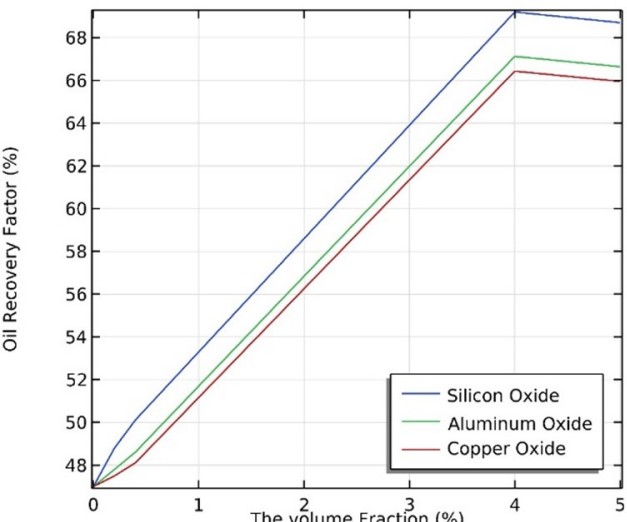

**Figure 16.** The effect of nanoparticle VFs on ORF.

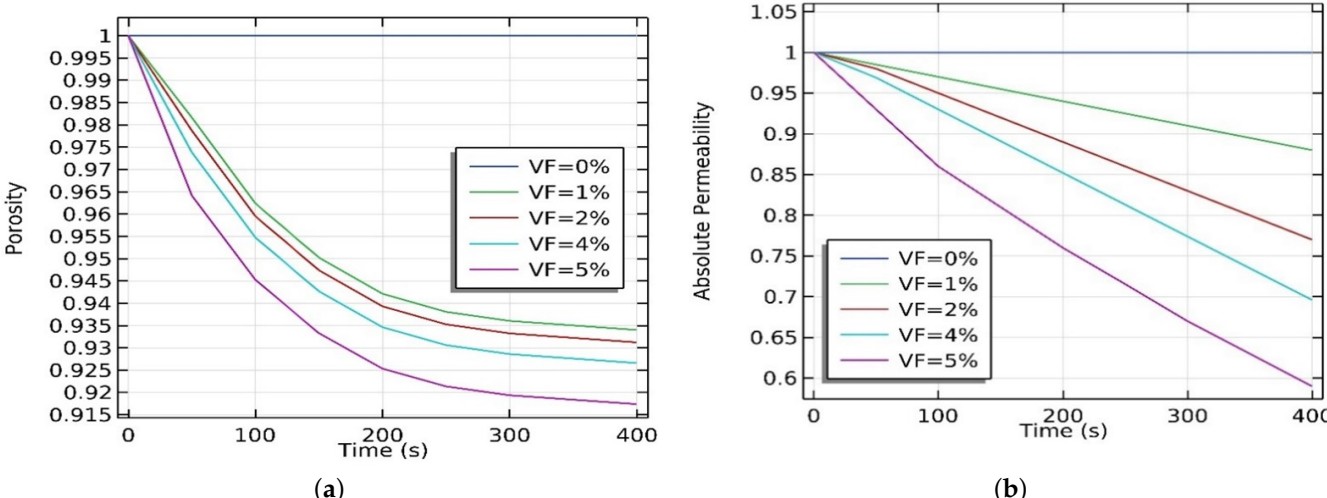

**Figure 17.** (**a**) The influence of nanoparticle VFs on porosity and (**b**) permeability.

*3.10. The Effect of Ideal NP SiO₂ on ORF at Different Inlet Fluid Temperatures and Volume Fractions*

After studying the effects of NFs (water-SiO₂, water-Al₂O₃, and water-CuO) on ORF, it was determined that the best candidate is water-SiO₂ for increasing ORF. After that, the effect of VF and temperature inlet of water-SiO₂ on ORF was tested to determine the best VF and temperature inlet for increasing ORF. It can be observed that the NP SiO₂ influenced the viscosity due to its TPs. As shown in Figure 18a,b, the ORF rose when the NF inlet temperature increased from 293.15 K to 353.15 K. Because the NFs increase the heat transfer efficiency of the NF to the oil inside the reservoir, it becomes lighter and needs less energy to flow out, resulting in more oil production. There are two reasons for this: first, the temperature that transfers through the NF to the oil lowers its density and viscosity, thus requiring less energy to move outside the PM; and second, it increases the temperature of the inlet fluid, causing an increase in the RP of NF and simultaneously decreasing the RP of oil, resulting in more oil production.

At vol. 0.5%, it can be deduced that there is a small improvement in the viscosity of oil and the NF (water-SiO₂), and the oil production increased in this case by 2%, 4%, and 5.5% at 293.15, 353.15, and 403.15 K, respectively. In the volume fractions of 1%, 2%, as shown in Figure 18a, and 3%, as shown in Figure 18b, SiO₂ caused an increase in the oil and the NF (water-SiO₂) viscosity to 8.188%, 7.3%, and 6.5%, and 12%, 22%, and 32%, respectively. As a result, the ORF increased to 4%, 9%, and 12% at 293.15 K; at 353.15 K increased to 8%, 14%, and 19%; and at 403.15 K improved to 10%, 18%, and 19.5%. At vol. 0.4%, SiO₂ decreased the oil's viscosity by 2.56% and increased the NF (water-SiO₂) by 37%. This means that NPs SiO₂, in this case, increases oil production by reducing oil viscosity when used with water as a base fluid. As shown in Figure 18b, at vol. 4%, the oil production increased by 20% and 37% and decreased to 7% at 293.15, 353.15, and 403.15 K, respectively. Finally, at vol. 5%, the oil production decreased by 17%, 30%, and 6% at 293.15, 353.15, and 403.15 K, respectively. Two main observations may be deduced from Figure 18a,b. The highest ORF of 37% is obtained at 353.15 K when SiO₂ is used at a VF of 4%, while the lowest recovery is obtained when vol. 5% is used at 403.15 K. So, the ORF increased when the temperature reached 353.15 K and then declined at 403.15 K when the VF is over 3%. The second observation is that the faster and higher ORF is obtained at the optimum temperature of (353.15 K) in this work when SiO₂ is used, which may reflect oil mobility. The optimum VF is 4%, where the RP of NF and oil increased and decreased, respectively, as shown in Figure 14a,b.

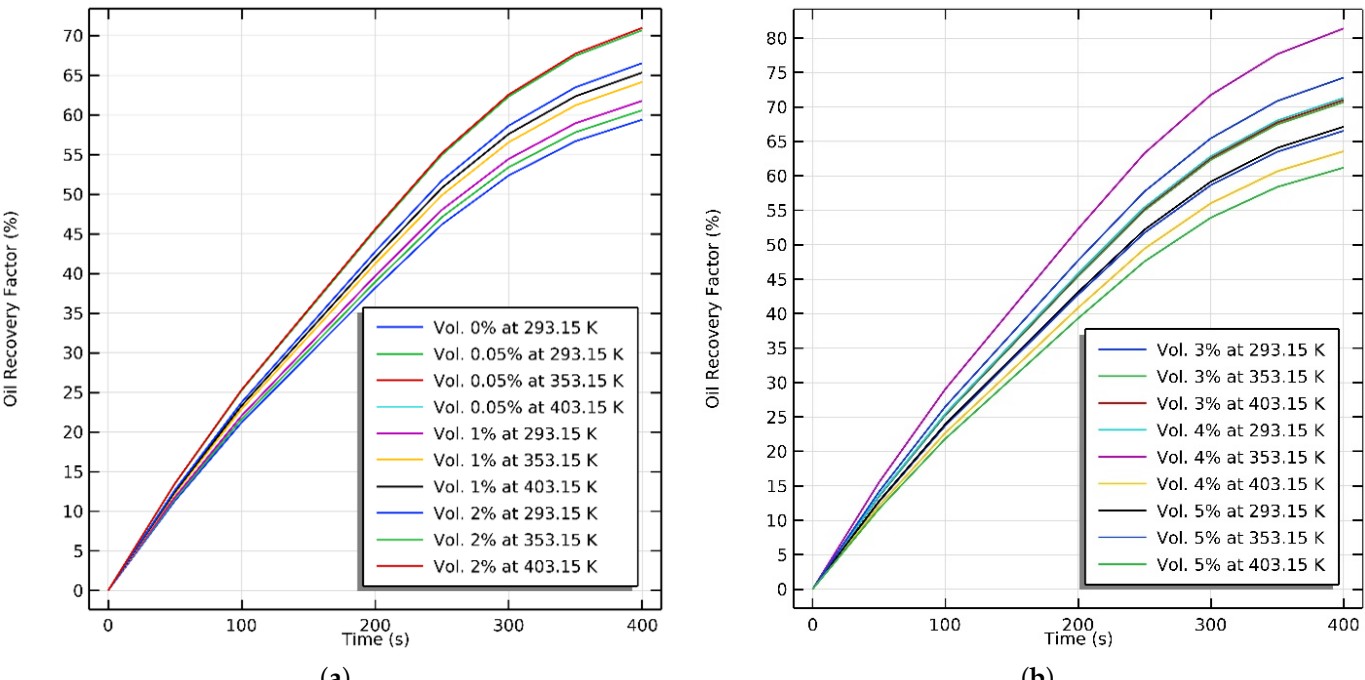

**Figure 18.** (**a**) The different inlet fluid temperatures effect of ideal NP SiO$_2$ on ORF at VFs (0.05, 1, 2%) and (**b**) At VFs (3, 4, 5%).

## 4. Conclusions

To sum up, the injection of NF into a three-dimensional oil-filled three-dimensional PM has been modeled. The governing equations of Darcy's law, mass conservation, concentration, and energy, have been solved using FEM code in COMSOL Multiphysics software. NPs thermophysical properties were studied for their impact on the base fluid and oil. The different types of NPs have been investigated in the present study, where the model has been run many times for SiO$_2$, Al$_2$O$_3$, and CuO to determine the NP with efficient ORF. To boost the ORF, the model was run with water as the base fluid six times to find the best VF (0.05, 0.1, 0.2, 0.3, 0.4, and 0.5%) of the best nanoparticle. Lastly, the NF inlet temperature has been modeled over the range of (293.15–403.15) to find the best inlet temperature for the best VF and NP for EOR. The results can be summarized as follows:

- The NP SiO$_2$ with TPs, lowest density, and highest thermal capacity is better than the other NPs and causes more oil production.
- After simulation and comparing the three different NPs, the results showed that SiO$_2$ is better than Al$_2$O$_3$ and CuO for EOR, in good agreement with the results from the present study and the findings from the literature, and 18% more oil production than the water flooding.
- The optimum NP (silicon oxide) and the temperature of 353.15 K increased the oil production by 25% more than water flooding.
- The best inlet temperature of injected NF, which fits the ideal NP and optimum VF, is 353 K, where the oil production increased to 37% more than water flooding.

**Author Contributions:** A.A.-Y. wrote the manuscript's main text, and A.A.-Y., D.L.C.C., H.S. and M.S.M. managed the project (methodology, software, validation, formal analysis, and data maintenance). D.L.C.C., H.S. and M.S.M. supervised the project, and H.S. funded the project. M.Z., Y.A., A.A.H.S. and A.H. reviewed the manuscript. All authors read and agreed to the published version of the manuscript.

**Funding:** This The authors thank the Department of Fundamental and Applied Science at Universiti Teknologi PETRONAS (UTP) for their assistance and support. This project was funded by grant cost center 015LC0-272 from Yayasan YUTP.

**Institutional Review Board Statement:** Not applicable.

**Informed Consent Statement:** Not applicable.

**Data Availability Statement:** The data presented in this study are available from the corresponding author upon reasonable request.

**Conflicts of Interest:** The authors declare no conflict of interest.

## Nomenclature

| | |
|---|---|
| $\varepsilon$ | Porosity |
| $D_c$ | Diffusion coefficient |
| $\rho_{nf}$ | Nanofluid density |
| $C$ | Nanoparticle concentration |
| $\mu_{nf}$ | Nanofluid viscosity |
| $R_i$ | The loss rate of the nanoparticles |
| $\boldsymbol{u}$ | Flow velocity |
| $k$ | Thermal conductivity coefficient |
| $K$ | Absolute permeability |
| $T_{in}$ | The temperature of inlet nanofluid |
| $p_{nf}$ | Pressure of the nanofluid |
| $\lambda$ | Pore size distribution index |
| $S_{nf}$ | nanofluid saturation |
| $B_c$ | The square root of permeability |
| $S_o$ | Oil saturation |
| $C_p$ | Specific heat capacity |
| $\boldsymbol{u}_{nf}$ | Input velocity of the inlet fluid |
| VF | Volume fraction |
| $P_c$ | Capillary pressure |
| $S_e$ | Effective saturation |
| $S_{rnf}$ | Residual saturation of nanofluid |
| $S_{ro}$ | Oil residual saturation |

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
