# Peer review of "Optimum Volume Fraction and Inlet Temperature of an Ideal Nanoparticle for Enhanced Oil Recovery by Nanofluid Flooding in a Porous Medium"

_processes, doi:10.3390/pr11020401_

Round 1

Reviewer 1 Report

The Present reviewer believes that the subject of the submitted MS is worthy of investigation and can be published after some corrections:

1.       The main question: How the optimum concentration of nanoparticle is calculated? Which optimization method has been utilized?

2.       The quality of the figures is not suitable, please change them and use the high quality figures. Especially figures: 8,9,4a,2,1

3.       It would be better to reorder the sections. Section 2-5 should be reorder to 2-3 (before grid independency), also the section 2-4 must be move to results section.

4.       Add a nomenclatures to make the MS more readable.

5.       The introduction section can be improved by adding some related papers:

10.4208/cicp.OA-2020-0001

https://doi.org/10.1016/j.ijheatmasstransfer.2018.12.060

Author Response

Response to Reviewer 1 Comments

The authors would like to thank the reviewer for the valuable comments and suggestions that contribute to improving our manuscript. Amendments were provided accordingly, and all modifications were highlighted red with track changes in the revised version of the manuscript. Response to the comments is given below.

Point 1: The main question: How the optimum concentration of nanoparticle is calculated? Which optimization method has been utilized?

Response 1: Thank you. The numerical simulation method has been used to determine the optimal concentration of nanoparticles in water for enhanced oil recovery. Computational fluid dynamics (CFD) uses mathematical models to predict the flow of fluids and the behavior of nanoparticles in porous media. In our work, we have used a multiphase flow in a porous media model.

The multiphase flow is a mathematical model that describes the flow of two immiscible fluids (nanofluid and oil) in porous media. It is used to predict the behavior of fluids in an oil reservoir during enhanced oil recovery operations.

The model uses the concept of relative permeability, which describes the effectiveness of each fluid in flowing through the porous media. The relative permeability of oil is generally lower than water's, meaning that oil flows more slowly through the rock than water.

To find the optimum concentration of nanoparticles in water for EOR, the two-phase Darcy law model, which is a multiphase flow model consisting just of two fluids, can be used to predict the relative permeability of oil and water in the presence of different concentrations of nanoparticles. By varying the concentration of nanoparticles, the model can predict how the relative permeability of oil and water changes and the effect on the recovery rate of oil.

The optimum concentration of nanoparticles is then found by comparing the recovery rate predicted by the model for different concentrations of nanoparticles and selecting the concentration that results in the highest recovery rate.

It is important to note that the optimum concentration of nanoparticles will depend on the specific reservoir and the type of nanoparticles being used. There are other factors, such as rock properties, wettability, interfacial tension, and viscosity, that might affect the relative permeability and oil recovery.

Point 2: The quality of the figures is not suitable, please change them and use the high-quality figures. Especially figures: 8,9,4a,2,1.

Response 2: Thank you. The quality of the figures has been improved as suggested (please see the revised manuscript).

Point 3:  It would be better to reorder the sections. Section 2-5 should be reorder to 2-3 (before grid independency), also the section 2-4 must be move to results section.

Response 3: Thank you for your valuable comment. Section 2-5 has been reordered to 2-3, and section 2-4 has been moved to the results section (please see the revised manuscript). 

Point 4: Add a nomenclature to make the MS more readable.

Response 4: Thank you. The nomenclature has been added in the revised manuscript (please see line 175)  

Point 5: The introduction section can be improved by adding some related papers:

10.4208/cicp.OA-2020-0001

https://doi.org/10.1016/j.ijheatmasstransfer.2018.12.060

Response 5:  Thank you. The papers have been added in the introduction section (please see line 125).

Reviewer 2 Report

The article "Optimum volume fraction and inlet temperature of an ideal nanoparticle for enhanced oil recovery bu nanofluid flooding in a porous media" presents relevant results related to flow in oil industry.

The manuscript is very well written. Only small issues should be considerer before the final version.

1 INTRODUCTION

a) I suggest to include in your review references to analytical approaches to deal with oil flows in porous media as performed in: https://doi.org/10.1016/j.petrol.2015.07.025

2 NUMERICAL IMPLEMENTATION

a) Figure 1 has a low resolution. It should be improved.

b) Sections 2.5.4 and 2.5.5 should include referentes for the general modeling/theory of the nanoparticle flow

GENERAL ASPECTS

a) I suggest to include as suplemmentary material the input files of the software used for reproducibility purposes.

Author Response

Response to Reviewer 2 Comments

The authors would like to thank the reviewer for the valuable comments and suggestions that contribute to improving our manuscript. Amendments were provided accordingly, and all modifications were highlighted in red track change in the revised version of the manuscript. Response to the comments is given below.

1 INTRODUCTION

  1. a) I suggest to include in your review references to analytical approaches to deal with oil flows in porous media as performed in: https://doi.org/10.1016/j.petrol.2015.07.025

Response 1: Thank you for your valuable suggestion. The analytical approaches have been included in the manuscript's introduction (please see lines 83–101 in the revised manuscript).

2 NUMERICAL IMPLEMENTATION

  1. a) Figure 1 has a low resolution. It should be improved.

Response 2: Thank you. Figure 1 has been improved (please see Figure 1).

  1. b) Sections 2.5.4 and 2.5.5 should include referentes for the general modeling/theory of the nanoparticle flow.

Response 3: Thank you. The references have been included in Sections 2.5.4 and 2.5.5 (please see lines 270, 274 and 298 in the revised manuscript)   

GENERAL ASPECTS

  1. a) I suggest to include as suplemmentary material the input files of the software used for reproducibility purposes.

Response 4: Thank you for your valuable suggestion.  The manuscript consists of 30 pages, including supplementary material would make the manuscript huge. However, the input files of the software are available from the corresponding author upon reasonable request. 
